# Adaptation of a Scientific Decision Support System to the Productive Sector—A Case Study: MOPECO Irrigation Scheduling Model for Annual Crops

Alfonso Domínguez [1,*], José Antonio Martínez-López [2], Hacib Amami [3], Radhouan Nsiri [4], Fadi Karam [5] and Maroua Oueslati [6]

1 Centro Regional de Estudios del Agua (CREA), Universidad de Castilla La Mancha (UCLM), 02071 Albacete, Spain
2 Instituto Técnico Agronómico Provincial de Albacete (ITAP), 02071 Albacete, Spain; josea.martinez@uclm.es
3 National Institute for Research in Rural Engineering, Water and Forests (INRGREF), B.P. 10, Ariana 2080, Tunisia; hacib.amami@gmail.com
4 National Institute of Field Crops, Boussalem 8170, Tunisia; nsiriradouan@yahoo.fr
5 Faculty of Agronomy, Lebanese University, Beyrouth 1003, Lebanon
6 Euro-Mediterranean Information System on Know-How in the Water Sector (SEMIDE), 09560 Valbonne Sophia-Antipolis, France
* Correspondence: alfonso.dominguez@uclm.es

**Abstract:** Despite the great number of models developed in research projects, only a small percentage have been successfully transferred to the productive sector. The PRIMA programme supported by Horizon 2020, the European Union Framework Programme for Research and Innovation, aims to reverse this situation. The SUPROMED project funded by PRIMA sought to develop an online platform composed of several models adapted to the requirements of end users for increasing the economic and environmental sustainability of Mediterranean agricultural systems. MOPECO, in its research version, was designed to maximize the profitability of irrigated farms in water-scarce regions. A simplified version of this model (MOPECO irrigation scheduling) was included in the SUPROMED platform for improving irrigation efficiency, providing farmers with a useful irrigation scheduling software. This paper shows the approach to adapt and transfer MOPECO to the productive sector. The tool was validated in three different demosite areas across the Mediterranean, involving local stakeholders in the design, validation, and dissemination of the software. The simplified tool reached similar or higher yields than farmers using less water. Thus, the average water saved was around 16%, while the average yield increased around 10% in the plots located in the three demosites of the project (Eastern Mancha in Spain, Bekaa valley in Lebanon, and Sidi Bouzid in Tunisia). This fact decreased the water footprint and increased the profitability of farms. The high applicability of the tool has generated interest among many technicians, farmers, and advisory enterprises. Furthermore, regional and national governmental extension services have shown interest in spreading the use of the tool across their territories, validating the methodology used for adapting and transferring a scientific model to the productive sector.

**Keywords:** stakeholders; sustainability; climate change; water scarcity; modelling; transference

## 1. Introduction

Although the scientific sector receives significant funding for the development of methodologies and tools intended for solving real problems, most of the models developed by research projects rarely get to be used by the productive sector. The main causes of this situation are the difficulties non-scientific or non-expert users experience in managing typically complex models, the lack of adaptation of the model to the real needs of the end users, and the impossibility of their obtaining some of the data required by the model [1]. Solving this situation has become a priority for public administrations supporting these

research activities, and for that reason, they demand a more effective transference of these outputs to the different stakeholders [2,3].

The European Union research programmes related to agriculture receive 2.2% of the EAFRD budget (European Agricultural Fund for Rural Development) [4], which is a considerable amount taking into account the agricultural sector receives the highest portion of the European budget (32% is destined to the Common Agriculture Policy "CAP") [5]. In 2020, European agriculture represented 1.3% of the GDP (Gross Domestic Product) [6], generating 9.48 million of jobs, and helped avoid the rural exodus and safeguarding food security in the EU.

The agricultural sector of Southern European countries, like others located in the rest of the Mediterranean basin, are obliged to deal with different threats, such as water scarcity, that impact on yields. This is worsened by the occurrence of drought periods that may increase with global warming, as well as low harvest prices and high input costs that reduce the profitability of the farms [7]. This highly variable situation causes insecurity across the productive sector, which needs to ensure the economic sustainability of its farms and is demanding more tools and methodologies to help farmers and technicians take the right decisions. With the aim of improving the resilience of Mediterranean agroecosystems, the PRIMA programme (Partnership for Research and Innovation in the Mediterranean Area) [8] promotes applied research projects in most Mediterranean countries.

To cope with this situation, many DSS (Decision Support Systems) have been developed for the farming sector [9–11], which, in most of the cases, require many input variables and parameter values that are not easily available to end users and involve advanced levels of training [12]. Dealing with these variables is arguably the main handicap when transferring a research model to end users, because they may initially feel overwhelmed. Other restrictions to be solved are to encourage end users to trust the results offered by the model and see that they really respond to their needs.

Consequently, several projects have been developed to adapt and transfer existing models and techniques used in the research community to the productive sector. From 2019 to 2022, PRIMA funded the SUPROMED (sustainable production in water-limited environments of Mediterranean agroecosystems) project, which aimed to improve the economic and environmental sustainability of Mediterranean farming systems. Thus, 10 models and methodologies adapted to end users and validated in three demosites areas were transferred to the productive sector through an online platform available on the project website (www.supromed.eu (accessed on 20 December 2022)). The models and tools were developed by some of the ten partners participating in the project, who come from five different nations (Spain, France, Greece, Lebanon, and Tunisia).

MOPECO (model for the economic optimization of irrigation water use at farm level) [13,14] was one of the models considered to be simplified and adapted to farmers and technicians for its inclusion in the SUPROMED platform, in order to allow producers to use a robust tool that has been validated under scientific standards. By making optimal use of the available irrigation water and cultivable area [10] MOPECO maximizes the profitability of irrigated farms in water-scarce arid and semi-arid regions. This model can be used in a large number of irrigable locations and has been calibrated for several crops [15–23]. Among other options, MOPECO generates an irrigation schedule taking into account the total amount of available irrigation water and complex concepts, such as the effect of irrigation uniformity [10], the electrical conductivity of irrigation water [15], and the differing sensitivity to water deficit at different crop growth stages [24]. For researchers, MOPECO, like other crop simulation models [12], needs a high number of variables and crop parameters that would make their widespread use by the productive sector impossible.

The main aim of this work was to transform a tool for scientific use into one for end users to achieve an efficient irrigation scheduling at farm level. In this way, the partial objectives of this research were: (1) to generate a simplified version of the irrigation scheduling module of MOPECO model adapted to farmers and technicians; (2) to validate the tool in the three pilot areas of SUPROMED Project (Spain, Tunisia, and Lebanon); (3) to

involve stakeholders in the development and validation of the tool; and (4) to transfer the tool to the productive sector.

## 2. Materials and Methods

### 2.1. Site Description

The methodologies and models included in SUPROMED were tested in three demosites across the Mediterranean area (Figure 1).

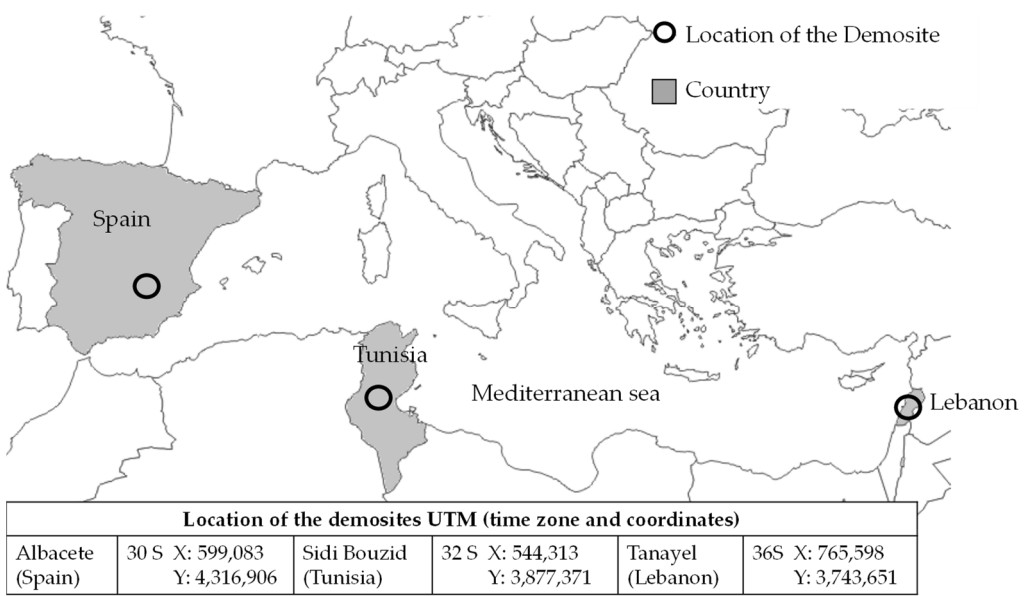

| Location of the demosites UTM (time zone and coordinates) | | | | | |
|---|---|---|---|---|---|
| Albacete (Spain) | 30 S X: 599,083 Y: 4,316,906 | Sidi Bouzid (Tunisia) | 32 S X: 544,313 Y: 3,877,371 | Tanayel (Lebanon) | 36S X: 765,598 Y: 3,743,651 |

**Figure 1.** Diagram of the location of the three demosites in the Mediterranean basin.

### 2.1.1. Spanish Demosite

In Spain, the demosite is located in the hydrogeological unit "Eastern Mancha" (HUEM), which occupies an area of 8500 km$^2$ and supplies water to more than 120,000 ha of irrigated land. In this area, 95% of irrigation systems are pressurized (mainly sprinkler and surface drip) with an average annual water allocation of 4000 m$^3$ha$^{-1}$. It is a semi-arid area where the average annual precipitation (350 mm) is distributed from September to June. The most common crops in the area are wine grapes, cereals (barley, wheat, and maize), garlic, and onion. The main problems in this area are the imbalance between water supply and demand, because 90% of the water used in the area is groundwater, increasing the risk of overexploitation of the aquifer [25]. Another important problem is the low animal production and the decrease in agricultural profitability due to the low sale price of harvests in combination with the high input costs and the low availability of irrigation water.

### 2.1.2. Lebanese Demosite

In Lebanon, the demosite is located in the South Bekaa valley, and is a part of South Bekaa Irrigation Scheme (SBIS), with an irrigated area of about 21,500 ha. Due to economic constraints, only a pilot area of 2000 ha is currently equipped with a pressurized irrigation network. The rest of the land is being irrigated through deep wells. SBIS has a semi-arid climate where an average precipitation of 650 mm is recorded from October to May. The most common crops are wheat and other winter cereals, potato, winter legumes and fruit trees, olives, and vineyards. The main problem in this area is the farmers' lack of knowledge of how to conduct appropriate irrigation scheduling and the lack of technical assistance from both the public and private sectors. The poor animal production system is another important problem.

### 2.1.3. Tunisian Demosite

In Tunisia, the demosite is located in Sidi Bouzid area, with an irrigated area of 50,000 ha, where 88% of the land belongs to farmers and the rest is managed by the government. This region is classified as arid, with an average annual precipitation of 250 mm. The main crops in the region include vegetables, fodder, and cereals. Tree crops include olive, pistachios, and almond. Most of the farms are equipped with pressurized irrigation systems using groundwater. Consequently, one the most important problems in this area is the overexploitation of groundwater, which causes water shortages and an increase in pumping costs. Other important problems are the lack of technical assistance and a poor animal production system.

### 2.2. Description of the MOPECO Model

The aim of MOPECO is to maximize gross margin (GM) through the efficient use of irrigation water and available irrigable land. A set of data (Figure 2) is required for the simulation of the optimal "Yield vs. Total Net Water" (Y vs. TWN) function of each crop under the climatic conditions of a certain year. In this function, TWN = net irrigation (IN) + effective rainfall (Pe). To obtain Y vs. TWN, the model simulates a range of deficit irrigation schedules using the optimized regulated deficit irrigation (ORDI) methodology, considering the effects of irrigation uniformity and electrical conductivity of water on yield. The Y vs. TWN function is translated into "Yield vs. Total Gross Water" (Y vs. TWG), where TWG = gross irrigation (IG) + Pe, to include the application efficiency of the irrigation system. The GM vs. TWG function is then calculated using economic data on the crop. Finally, the model calculates the optimal distribution of crops that fulfil the restrictions imposed by the user (crops, irrigable area, and available amount of water for irrigation) (Figure 2).

Daily actual evapotranspiration $ET_a$ is derived using the equation proposed in [26], which calls for a daily soil water balance, whereas daily maximum evapotranspiration $ET_m$ is produced by multiplying daily crop coefficient ($K_c$) by daily reference evapotranspiration ($ET_o$) [27]. The calculation from [28] is then used to determine the percentage of total available soil water (TAW) (mm) that the crop may draw without experiencing water stress (dimensionless).

As a result, there are three variables in the calibration of Y–TWN: (a) potential yield ($Y_m$), which can be obtained through field trials; (b) $K_c$; and (c) the crop yield response factor for each growing stage [29] ($K_y$). The length of the different crop growth stages is required for the simulation of crop growth cycle in terms of Cumulative Growing Degree Days (CGDD) calculated according to [30]. The phenological scale used is the Biologische Bundesanstalt, Bundessortenamt und CHemische Industrie (BBCH) [31], which assigns a number to each stage.

Thus, the simplified tool will use the irrigation scheduling module of MOPECO for researchers, which is based on FAO-56 methodology [15,26,32], and only calculate the irrigation scheduling for one point of the Y vs. TWN function, which corresponds to the yield under no water deficit conditions.

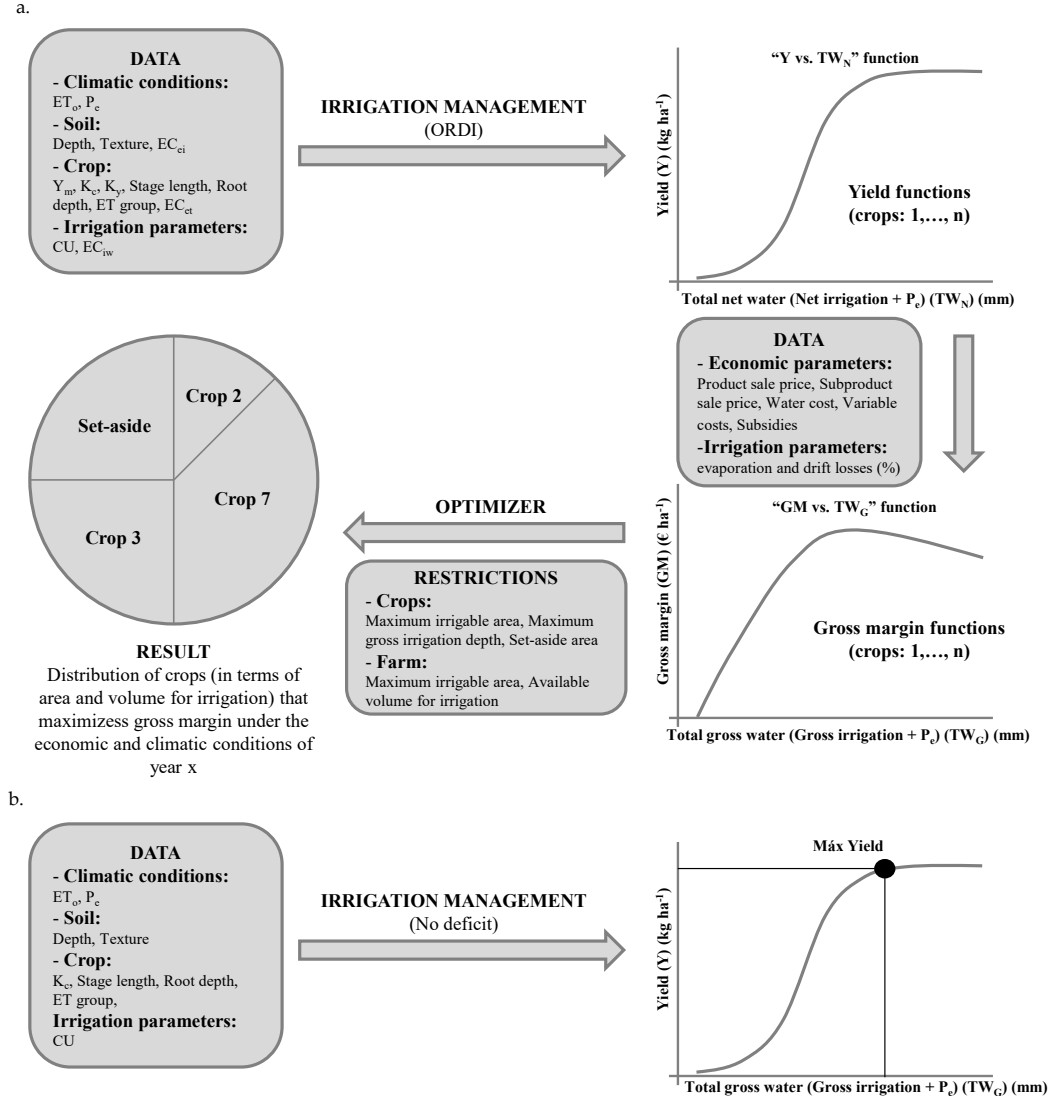

**Figure 2.** (**a**) Flow chart of MOPECO for researchers [17]; (**b**) flow chart of the simplified version of MOPECO: ETo: reference evapotranspiration; Pe: effective precipitation; $EC_{ei}$: electric conductivity of the soil; $Y_m$: maximum yield; $K_c$: crop coefficient; $K_y$: crop yield response factor; CU: uniformity coefficient; $EC_{iw}$: electric conductivity of irrigation water.

### 2.3. Methodology to Develop a Model That Can Be Successfully Transferred to the Productive Sector

Following the methodology developed by [33], the most important aspects to consider when developing a model to be successfully transferred to the productive sector are:

a.  Identifying the problems and the owners of these problems;
b.  Considering how, and by how much, the proposed model solves the problems of end users compared to other alternatives;
c.  Defining the use of the model and its customers. The type of use can be:

- Direct: as a background for further research and development, provision of a service, sales of a product/process, adoption in a standard;
- Indirect: facilitating use to third parties (transfer of results, licensing, creation of a spin-off).

d.  Identifying the key exploitation result (KER):

- The KER must respond to the needs of specific target groups and match the commitment of a project partner;

- It should be easily understandable for end users and be described in such a way that others can visualize it;
- A KER is not only a product/service; it could also be scientific knowledge, a new policy, a demonstrator, etc.

e. The terminology used in the model and the information required to run it must be understood and easily available for end users;

f. Validating the model outside the partnership. Once the problem and the problem owners are identified and the model is adapted to them, it is important to validate the model under the actual conditions in which the model will be used;

g. To transfer the model to end users. During the transfer process of the model, it is key not to forget the validation process because it is the first step of the transfer process, offering a way to reach out to "early adopters". It is also important to consider the channels to be used to reach them;

h. Organizing the team for implementation, identifying the key roles and profiles needed (researchers, informatics, technicians, etc.), involving people with experience in "going to market";

i. Defining the follow-up activities for the period after the end of the project:

- Planning, organizing, and ensuring the follow-up activities. Adopting a solution always requires activities to be carried out after the project;
- Responsibilities on follow-up activities;
- Resources needed for follow-up activities and ensuring the future use of the model.

### 2.3.1. Identifying Problems Affecting Irrigated Farms

The main problems affecting irrigated farms in the Mediterranean basin are well known:

- Low availability of water for irrigation worsened by drought periods, which may cause overexploitation of water resources [34];
- Low profitability of farms caused by low harvest prices and the high costs of inputs such as energy and fertilizers [35];
- Low yield productivity of rainfed farms;
- Global warming may exacerbate the above problems [36].

Other important problems that depend on the area are:

- Aging of inhabitants in rural areas increasing the risk of rural depopulation in the future [37];
- Low assistance of public institutions to the sector;
- Lack of irrigation advisory services and/or useful models and tools adapted to the productive sector that may assist farmers in their decisions [38].

In order to determine the specific and general problems in the 3 demosite areas, a questionnaire developed by the research team was distributed among stakeholders. Moreover, the questionnaire was also used to obtain data on agricultural practices, use of decision support systems, willingness to pay for agricultural services, and questions related to socioeconomic characteristics of the farmers. Therefore, the responses in the questionnaire were used to design a tool capable of solving one or more of the problems described by stakeholders.

### 2.3.2. Considering How, and by How Much, the Proposed Model Solves the Problems of End Users Compared to Other Alternatives

Carrying out efficient irrigation scheduling is essential for improving water and energy use in irrigated agriculture [38]. The main methodologies used to determine irrigation scheduling may be summarized as the following [38]:

- The farmer's experience/perception of crop irrigation needs, which usually results in less-than-optimal irrigation scheduling (lack of water during some crop stages

and over-irrigation in others) and, hence, lower water productivity, production, and profits [39];

- Rational estimation of daily crop irrigation requirements using historical climate data that are daily updated during the irrigation season [40];
- Rational estimation of daily crop irrigation requirements based on climate data and on daily soil water balance [41,42];
- Models able to determine the irrigation requirements of crops, including climatic data, crop phenological development, and soil water balance. Such models include Aquacrop, CropSyst, and Isareg, among others [9,11,43];
- Estimation of daily irrigation needs using soil water data collected from soil moisture sensors [44,45];
- Estimation of irrigation needs using remote sensing [46,47];
- Irrigation needs estimated from plant water status monitoring, using sap flow sensors [48,49], trunk growth rate sensors [50], leaf water potential [51], or leaf turgor pressure sensors [52], among other methods;
- Irrigation advisory services using a combination of the former methodologies [38,53].

The efficiency in the use of irrigation water of the above methodologies is highly variable (from poor to excellent), with the higher the efficiency, the greater the training and investment.

### 2.3.3. Defining the Use of the Model and Its Customers

The model should provide end users with practical and accurate information about how to implement efficient irrigation scheduling. The model must provide results for the entire growing season of the crops and be adapted to the training and data available for the users.

The main direct users of the tool will be farmers of low to high agronomic and technological training who could directly and independently use the tool as an irrigation scheduling DSS. In the same way, technicians working in irrigation advisory services or engineering consultants are also expected to be direct users.

Additionally, indirect users might be irrigation assessment companies that use the tool to advise landowners who wish to implement proper irrigation scheduling but are unable to use the tool because of low-level technological training or the large extension of their farms. Public administrations might also be interested in promoting the use of the tool to improve the productivity of agricultural systems and decrease the impact of agriculture on the environment.

### 2.3.4. Identifying the Key Exploitation Result (KER)

The main result is a software that can bolster the ability of farmers to increase the profitability of their farms through a more productive use of water, mainly under water-scarce conditions. It should offer useful recommendations to improve irrigation scheduling to achieve the potential yield of the crops, avoiding the saturation of the soil, thus decreasing percolation, fertilizer leaching, and diseases related to excess humidity. It should take advantage of the rainfall and reduce irrigation expenditure through a lower use of irrigation water and the energy costs associated with the extraction and pressurization of this water. Therefore, the tool should improve efficiency in the use of water resources, decrease the impact on natural resources, and increase the profitability of Mediterranean farming systems, helping anchor population in rural areas.

### 2.3.5. The Terminology Used in the Model and the Information Required for Running Must Be Understood and Easily Available for End Users

Adapting a model designed for researchers to be implemented by end users is no simple task. Models for researchers typically consider a large number of variables and parameters to achieve highly accurate and robust results [9,11,43]. In the case of irrigation models used for simulating the development of crops, some of these variables can be



obtained easily (i.e., soil texture and depth, sowing date, etc.), but others are complex to understand (i.e., irrigation uniformity, growing-degree-days, allowable depletion level, etc.) or difficult to obtain (i.e., $ET_o$, $K_c$, threshold development temperatures, etc.). For this reason, it is necessary to analyse the level of difficulty of each variable and parameter required by the model, determine which may be requested from the user, and which must be simplified at the expense of decreasing the accuracy of the model within acceptable ranges.

In order to improve the level of adaptation of the tool to the requirements and training of end users, their active participation was encouraged in the SUPROMED project. It is worth noting that users can also contribute to the process with on-site information that may be unknown to the research team. Their point of view typically differs from that of researchers. Therefore, they help increase the diversity of criteria and create a more complete DSS [25]. Moreover, to improve the adaptation of the tool [54,55] questionnaires were used to collect the opinions of general end users, and personal meetings with selected farmers and technicians representative of the sector were conducted during the project.

### 2.3.6. Validating the Model Outside the Partnership

The validation of the model is required to meet two objectives. On the one hand, it is necessary to show that the results of the tool under actual management conditions are accurate, for which scientific methodologies must be used. On the other, the results of the validation can be used to demonstrate to farmers and technicians how the use of the tool can improve the profitability of the farm by a more efficient use of water compared with their traditional management (no use of the tool). For the validation process, it was proposed that we monitor several commercial plots, with some of them being managed by the research team.

To fulfil the first objective, the tool should be validated for the main crops in the area. In the case of Spain, MOPECO was previously calibrated in the area for barley [56], maize [16,17], onion [18], garlic [19,57], melon [20], and potato [23]. In any event, other validated crops were included in the tool, using the parameters published by other researchers (i.e., oat, wheat, and alfalfa [58]). This second option was used for the crops monitored in Lebanon and Tunisia, due to only potato having been previously calibrated for MOPECO in the area [15]. It should be noted that the model established in Tunisia for determining irrigation scheduling of annual crops was IREY [59], which requires similar parameters to those demanded by MOPECO.

For the validation at farm level and to achieve a proper comparison with the results obtained by traditional management, one highly trained and high producing farmer per each crop was selected as "leader" farmer (LEA). This farmer allowed SUPROMED partners (SUP) to manage a portion of their farm, using the tool and other methodologies proposed by the project. Thus, two plots of similar characteristics on their farms were monitored during the first year: one managed as usual by the LEA, and the other by SUP. The monitoring of the plots involved analysing the soil characteristics for proper fertilization and irrigation scheduling, installing soil moisture sensors, evaluating the irrigation system and installation of pressure transducers and flowmeters, and the weekly monitoring of the crop development and the labours carried out in the plot. In order to increase the number of monitored plots and determine the real impact of establishing the tool in the area, other lower trained producers, termed here average farmers (AVE), were also involved in the monitored tasks. The objective was to check if the model was running properly in different conditions and to show AVE how they could improve their management by using it. Thus, they would act as early adopters and could help us in disseminating the tool among other farmers. A more detailed description of this process was described in [54,55].

### 2.3.7. Transferring the Model to End Users

The comparison between traditional management and that proposed by the simplified MOPECO tool was considered the first step for convincing farmers of the effectiveness of the model. This was the beginning of the feedback process between the farmers and the

research team, which took place during the second year of the project when the farmers started to use the software in their farms, motivated by the results obtained by the research team in the previous year [54,55].

In addition, training courses and technical meetings were organized to show these results to other farmers and extend the use of the tool to the productive sector. Three sets of training in each demosite involving at least 50 farmers were planned. Workshops linked to demosites targeted end users, local agriculture authorities, NGOs, and national agriculture and water authorities. Sessions were mainly dedicated to local/national stakeholders, provided in the national languages by the persons in charge of the demosite areas. In the case of participating speakers from other countries, English was simultaneously translated to the local language.

Other actions proposed to increase the transference of the model to end users were the development of tutorials for the use of the model, free access to the tool through a web site, and the appearance of news related to the tool on social media, such as Facebook, Twitter, and LinkedIn, and in the local press, radio, and TV enterprises.

### 2.3.8. Organizing the Team for Implementation

To implement the simplified irrigation scheduling model as a useful tool adapted to end users, it was necessary to form a multidisciplinary team composed of:

- Researchers: responsible for the scientific development of the model, its calibration and validation, as well as its adaptation and simplification;
- Technicians: mostly agricultural engineers, in charge of carrying out the process of validation and simplification of the model in cooperation with the research staff;
- Software developers: responsible for the development and layout of the application, creating a simple and attractive environment for end users;
- Early users: in this case, innovative farmers and agricultural engineers, who checked the proper operation of the tool on their farms, participated in the adaptation and simplification of the tool, and transmitted their experience to other farmers.

### 2.3.9. Defining the Follow-Up Activities for the Period after the End of the Project

After ending the SUPROMED project, transferring the simplified MOPECO model to end users entails:

- Maintaining the server where the irrigation scheduling application is hosted;
- Maintaining the network of weather stations that feed the model. This task involves maintaining the plot where the station is located, and cleaning and calibrating sensors and their connection to the application server. In the case of Castilla–La Mancha, the SIAR network is maintained by the Ministry of Agriculture;
- Updating the tool with new crops and varieties, which, in turn, involves carrying out research activities for their proper calibration;
- Developing new functions to improve the use and results of the tool;
- Disseminating the tool in other areas to involve new users and train them in its use.

## 3. Results and Discussion

### 3.1. Generating a Simplified Version of the Irrigation Scheduling Module of MOPECO Model Adapted to Farmers and Technicians

The simplified version of the MOPECO model was called "MOPECO irrigation scheduling" (*Servicio Integral de Asesoramiento al Regante: Programacion de Riegos* "SIARPR" in the version for Spanish farmers) and is available for free online https://crea.uclm.es/SIARPR/ (accessed on 2 January 2023) and in the app store (smartphone version).

According to the results of the questionnaires and personal interviews, end users in the three areas demand a tool that can achieve suitable irrigation scheduling without requiring investment in sensors or paying a great amount for the service. This response invalidates many of the actual ways to determine a suitable irrigation schedule. Thus, the most suitable option was to develop a free software that requires a low number of variables.

The simplified MOPECO tool decreased the total number of data the user is required to enter in the tool to 10 (4 more for advanced users) compared to 43 for the case of MOPECO for researchers (Table 1). Thus, all the parameters labeled as "Not Considered" were not included in the simplified version because they were not necessary for determining the irrigation scheduling (i.e., total area or potential yield), or because the methodology was simplified (i.e., the tool does not consider the effect of salinity nor calculate regulated deficit irrigation schedules).

- Climatic data: They are linked to the location of the plot, which is easily entered by the user through a GIS viewer (Figure 3). The application automatically selects the nearest weather station, from which it automatically collects the climatic data necessary for the calculation (Table 1). In any event, the model allows the user to select any of the 5 closest weather stations to the plot.

**Table 1.** Data required by the simplified "MOPECO irrigation scheduling" tool for farmers and their source.

| MOPECO for Researchers<br>Required Data | MOPECO<br>Irrigation Scheduling |
|---|---|
| **Farm** | |
| Location of the plot (coordinates) * | User |
| Total area (ha) | Not considered |
| Total available volume of irrigation water ($m^3$) | Not considered |
| **Climatic** | |
| Daily reference evapotranspiration "$ET_o$" (mm) | Weather station |
| Daily max. temperature (°C) | Weather station |
| Daily min. temperature (°C) | Weather station |
| Daily rainfall (mm) | Weather station/User ** |
| Effective rainfall (%) | Estimated |
| **Soil** | |
| Texture | User |
| Field capacity (mm $m^{-1}$) | Estimated |
| Wilting point (mm $m^{-1}$) | Estimated |
| Depth (m) | User |
| Stone content (%) | User |
| Initial soil moisture content (%) | User |
| Electrical conductivity of saturation extract "ECe" (dS $m^{-1}$) | Not considered |
| **Crop** | |
| Potential yield "Ym" (kg $ha^{-1}$) | Not considered |
| Yield response factor per development stage "Ky" | Not considered |
| Ky for the entire growing period "Kyg" | Not considered |
| Crop coefficient per development stage "Kc" | Calibrated |
| ET group for determining the soil depletion before stress | Calibrated |
| Cumulative growing degree-days "CGDD" (°C) | Calibrated |
| Lower developmental threshold temperature "$T_L$" (°C) | Calibrated |
| Upper developmental threshold temperature "$T_U$" (°C) | Calibrated |
| Root depth (m) | Calibrated |
| Sowing date (dd/mm/yyyy) | User |
| Max. ETa/ETm difference between consecutive stages | Not considered |
| Threshold above which yield is affected by salts "ECet" (dS $m^{-1}$) | Not considered |
| Rate of yield decrease due to salts "b" (% $dS^{-1}$ m) | Not considered |
| Harvest sale price (EUR $kg^{-1}$) | Not considered |
| Subproduct sale price (EUR $kg^{-1}$) | Not considered |
| Variable costs (EUR $ha^{-1}$) | Not considered |
| Subsidies (EUR $ha^{-1}$) | Not considered |
| Max. cultivable area (ha) | Not considered |
| Max. irrigation amount ($m^3$) | Not considered |

**Table 1.** *Cont.*

| MOPECO for Researchers Required Data | MOPECO Irrigation Scheduling |
|---|---|
| **Irrigation system** | |
| Max. interval between irrigation events (days) | User |
| Min. interval between irrigation events (days) | User |
| Max. irrigation depth supplied (mm) | User |
| Min. irrigation depth supplied (mm) | User |
| Coefficient of uniformity (%) | Integrated in efficiency |
| Efficiency (%) | Estimated/User ** |
| Readily available soil water refill level (%) | Recommended/User ** |
| Readily available soil water depletion level (%) | Recommended/User ** |
| Electrical conductivity of irrigation water "ECiw" (dS m$^{-1}$) | Not considered |
| Water cost (EUR m$^{-3}$) | Not considered |

Notes: * Not considered in the version for researchers; ** advanced user.

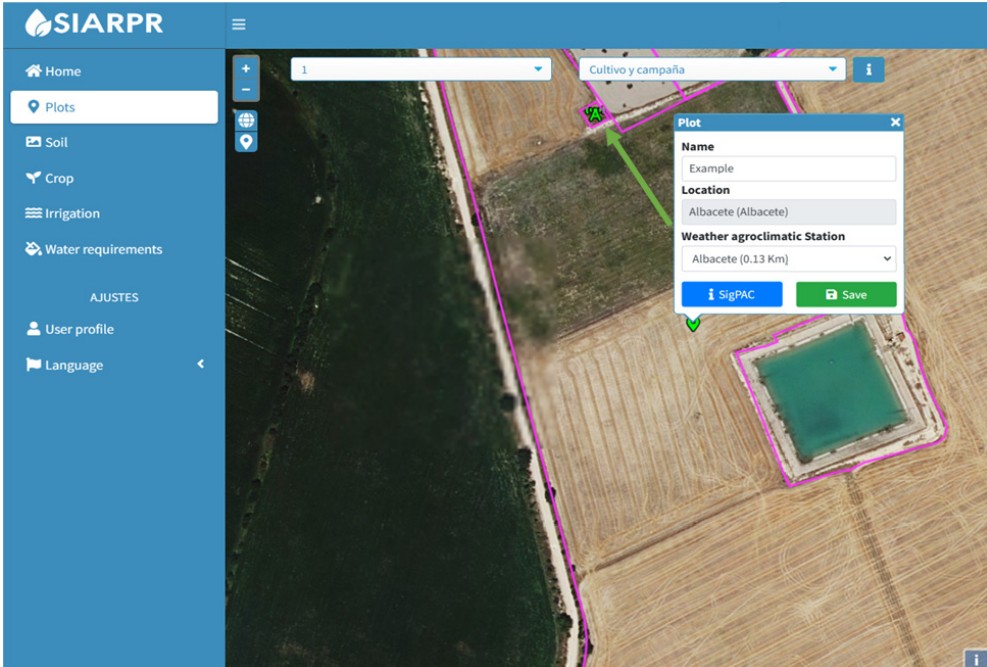

**Figure 3.** Location of the plot and selection of weather station.

The model uses three different climatic databases: the one from the closest weather station for determining the actual irrigation requirements of the crop from sowing up to the current date; the next seven days' irrigation requirements by using weather forecasting services available in the area (i.e., the national agency of meteorology (INM) in Spain [60]); and the estimated total crop cycle irrigation requirements by using the typical meteorological year (TMY) [19] calculated for the selected weather station. In the same way, farmers are recommended to install rain gauges in their plots in order to obtain more accurate rainfall data than those provided by the weather station (Figure 4).

- Soil data: According to the type of texture selected by the user, the software assigns certain average field capacity and wilting point values obtained from the bibliography. While establishing these values is difficult for farmers, determining the texture of their soils is easy via the soil analysis used to carry out for applying a proper fertilization (Figure 5). In the same way, they can easily estimate or measure the useful depth of their soils and the percentage in volume of stones.

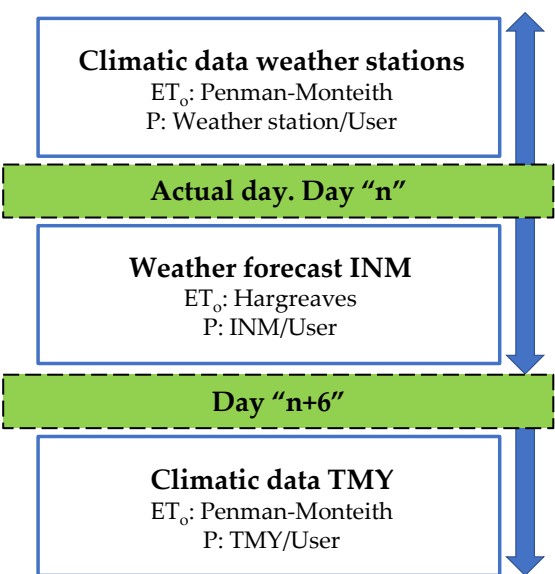

**Figure 4.** Climatic data used for the calculation of crop water requirements.

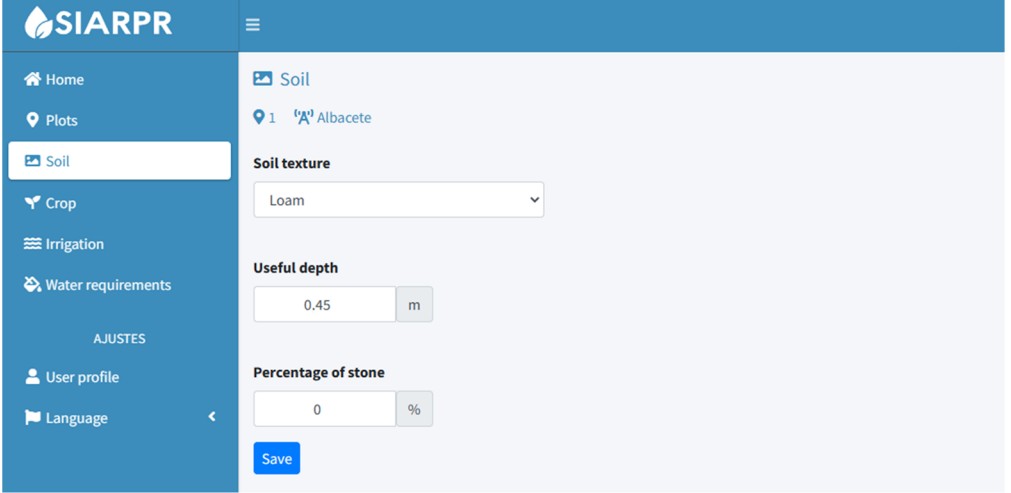

**Figure 5.** Soil data for the cultivated plot.

- Crop data: The user has to select the crop to be cultivated in the plot from a list and insert the sowing date. The rest of the parameters required to simulate the crop cycle (Table 1) were previously entered in the tool (not visible for users) by the research team in charge of calibrating the model in the area where the model is being used. It is recommended that these data come from research experiments carried out in the region. Evidently, crops not inserted in the tool by the research team cannot be simulated by the model. Automatically, the program simulates the total length of the crop cycle and of the main growing stages (those related to the $K_c$ progression). The duration of the stages can be modified by the user during the season to fit the estimated progression to that observed in the field. To facilitate the identification of the key phenological stages of the crop related to the change in $K_c$ values, some descriptive pictures are shown (Figure 6).

- Irrigation system data: The required values (Table 1) must be entered by the user since they are specific to each irrigation system. It is recommended to periodically carry out an evaluation of the irrigation system to obtain updated, accurate values. The most common values for the systems in the area are provided by default. Moreover, the "advanced irrigation settings" section (Figure 7) allows users to define the initial soil

moisture content on sowing day and determine the soil refill and depletion levels after an irrigation event for arable crops, which are, respectively, set at 75 and 50% of easily available water by default [61].

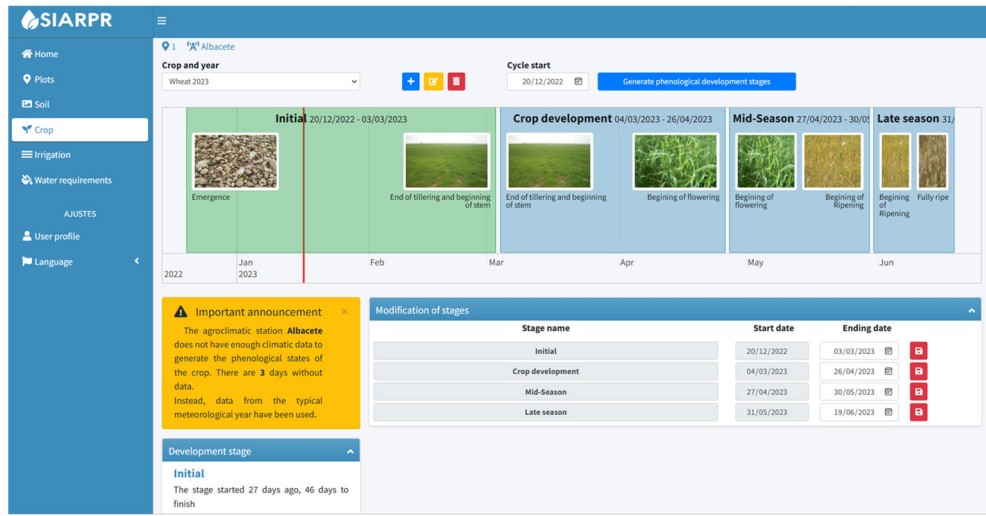

**Figure 6.** Crop data and theoretical phenological progression.

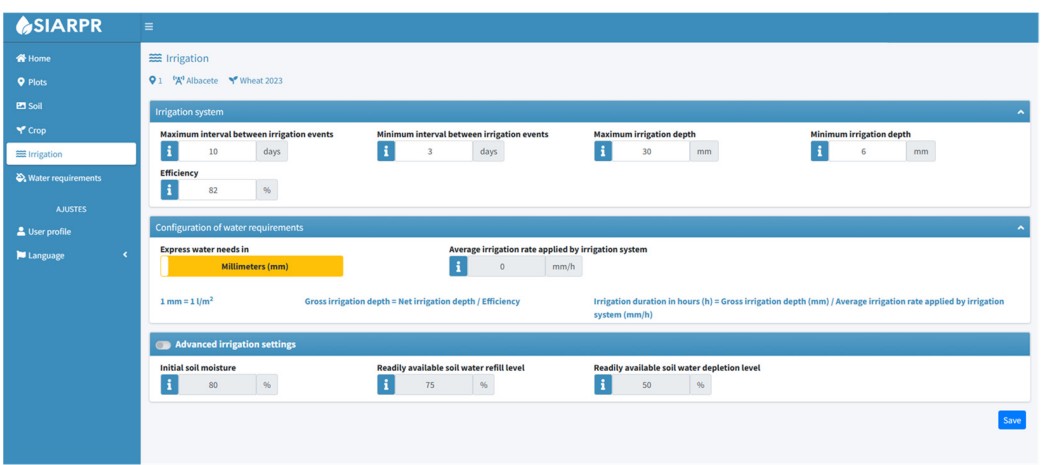

**Figure 7.** Irrigation and advanced irrigation settings.

Finally, the programme offers the irrigation scheduling from the moment of the simulation up to the end of the growing season as a result of the daily water balance in the soil, calculating the necessary amount of irrigation water and when to supply it to cover the water needs of the crop, avoiding water deficit and percolation. Thus, the aim is to maintain the total available soil water level (TAW) line (purple line) below the field capacity level represented by the top of the graph (TAW = 1) and over the maximum allowable depletion level (red area) (Figure 8). The TAW line moves between wilting point (TAW = 0) and field capacity (TAW = 1). The model saves the actual irrigation amounts supplied by the farmer (green dots) during the period before the day of the simulation (the user can modify the irrigation events proposed by the model, changing the colour from green to orange), as well as the rainfall data (blue triangles) entered by the user if appropriate (colour change from blue to red), with the irrigation requirements for the following seven days being the most important information for the user. The model daily updates the soil water balance when changing the values forecast to those actually registered by the weather station when available, as well as when the user modifies the amounts supplied by the irrigation system or the actual rainfall registered by their rain gauge.

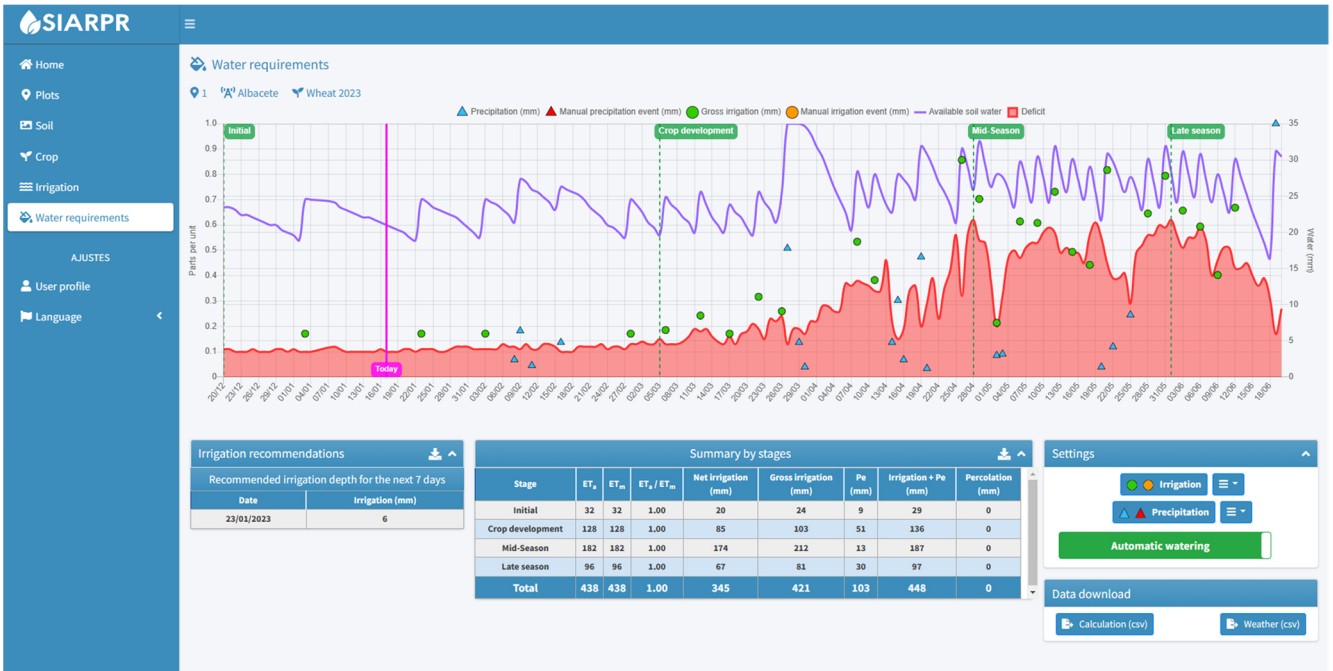

**Figure 8.** Calculation of soil water balance and the irrigation scheduling for a maize crop.

### 3.2. Validating the Tool in the Three Pilot Areas of SUPROMED Project

During the 3 campaigns over which the validation process described in Section 2.3.6. was carried out, 51 plots cultivated with 11 different crops belonging to 18 different farmers were monitored (Table 2). During the first campaign, at least 3 plots were monitored for each crop, involving at least 2 farmers (one leader and one average), except at the Lebanon demosite (only leader). Five weather stations were installed in the demonstration areas and 51 groups of sensors were installed to monitor soil moisture progression in the plots. During the second campaign, a preliminary version of the tool was transferred to various leader farmers interested in the use of MOPECO, motivated by the results obtained by the research team in the first campaign (Figure 9). Thanks to this process, the farmer was able to know, at a practical level, how to manage the tool assisted by the research team and, additionally, we obtained useful information for a proper adaptation of the tool to the productive sector [54,55].

**Table 2.** Crops monitored in the SUPROMED project.

| Crop | Demosite | Number of Monitored Plots | | | Total |
|---|---|---|---|---|---|
| | | 2019–2020 | 2020–2021 | 2021–2022 | |
| Barley | S | 5 [a,b,d] | 1 [c] | | 6 |
| Fodder Oats | S | 3 [a,b,d] | | | 3 |
| Grain Oats | S, T | 6 [a,b,d] | | | 6 |
| Garlic | S | 4 [a,b,d] | 3 [c,d] | | 7 |
| Alfalfa | S | 3 [a,b,d] | 1 [c] | | 4 |
| Wheat | L, T | 6 [a,b,d] | 4 [a,b] | | 10 |
| Potato | L | 2 [a,b] | 2 [a,b] | 2 [a,b] | 6 |
| Silage maize | L | 2 [a,b] | | | 2 |
| Onion | T | 3 [a,b,d] | | | 3 |
| Maize | S | | 2 [a,b] | | 2 |
| Sweet maize | S | | 2 [a,b] | | 2 |

Notes: S: Spain; L: Lebanon; T: Tunisia; [a]: plots managed by SUPROMED team; [b]: plots managed by leader farmers in their traditional way; [c]: plots managed by leader farmers using the tool; [d]: plots managed by average farmers.

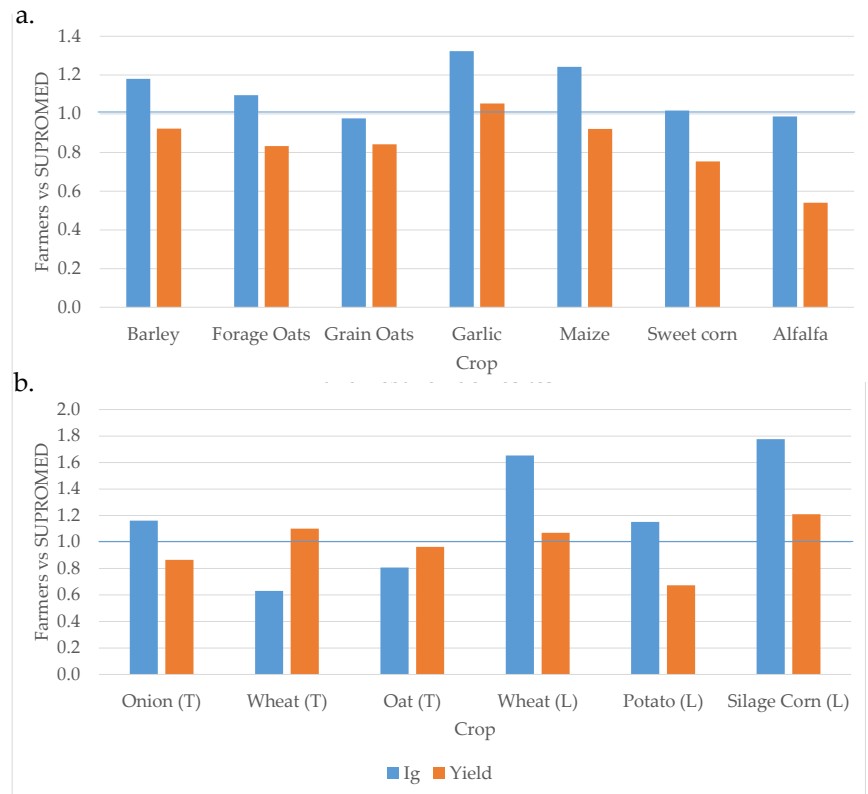

**Figure 9.** Farmers vs. SUPROMED management ratio for Yield and Ig (Gross irrigation) variables. *Y* axis: ratio farmers vs. SUPROMED calculated dividing the amounts of gross irrigation water supplied and yields obtained by farmers with regards SUPROMED management. *X* axis: monitored crop and location of the plot ((**a**) Spain; (**b**) Lebanon "L" and Tunisia "T").

The evaluation of 24 irrigation systems (Table 3) showed that the uniformity of most of the farms was around 73%, which can be considered acceptable [62]. Regarding the amount of water discharged by the systems, this ranged between 4.0 and 7 mm h$^{-1}$, which is the common dose in the area [63]. However, in some cases, the values obtained were significantly different to those used by farmers for determining the working time of the sprinklers or the speed of the centre pivot to supply the desired irrigation depth. The main cause of these differences was the age of the irrigation system, due to the actual running conditions becoming worse with time.

The irrigation management of the plots using the "MOPECO irrigation scheduling" tool (and other methodologies proposed by SUPROMED platform) allowed the research team to improve the yield obtained by farmers for most of the monitored crops, utilizing, in most cases, a lower amount of gross irrigation water (Ig). Thus, by calculating the ratio between Ig supplied by farmers with regards to SUPROMED team, and the yield (Y) ratio, SUPROMED team improved the results in most of the demonstration plots (Figure 9). This fact also improved most of the analysed key performance indicators "KPIs" (Table 4) (water productivity, irrigation water productivity, gross margin, water footprint, etc.) that were determined to analyse the impact of the SUPROMED tools in the farms [54,55]. Moreover, these results corroborated those obtained during the calibration process in Spain [19,56,64], and are also in line with the average irrigation recommendations made by the main irrigators association in the area [65]. In addition, the model reached a high level of fit between the simulated and measured progression of soil water content by using different types of moisture sensors [54–56], confirming that the model was properly calibrated for such crops in the study area.

**Table 3.** Results of irrigation system evaluations.

| Crop | Manager | Sprinkler Spacing (m × m) | Pressure (kPa) | Sprinkler Discharge (L h$^{-1}$) | Application Rate (mm h$^{-1}$) | DU (%) | CU (%) |
|------|---------|---------------------------|----------------|----------------------------------|-------------------------------|--------|--------|
| Barley | SUP * | 17.3 × 17.3 | 402.5 | 2053 | 6.9 | 75.7 | 85.9 |
| | LEA * | 17.3 × 17.3 | 358.8 | 1967 | 6.6 | 77.8 | 87.4 |
| | AVE 1 * | 17.3 × 16.8 | 366.4 | 2109 | 7.0 | 76.5 * | 86.7 * |
| | AVE 2 * | 17.3 × 17.3 | 354.4 | 1963 | 6.6 | 76.5 * | 86.7 * |
| | AVE 3 * | 17.5 × 17.5 | 403.0 | 2085 | 6.8 | 43.8 | 68.5 |
| | LEA$_{SUP}$ ** | 17.3 × 17.3 | 403.8 | 2003 | 6.7 | 79.4 | 85.9 |
| Oats | SUP * | 17.3 × 17.3 | 398 | 2083 | 6.9 | 75.7 | 85.9 |
| | LEA * | 17.3 × 17.3 | 398 | 2049 | 6.9 | 77.8 | 87.4 |
| | AVE [1] * | 17.3 × 17.3 [1] | 309 | 1839 [1] | 6.1 [1] | 76.5 * | 86.7 * |
| Garlic | SUP * | 17.3 × 17.3 | 404 | 2003 | 6.7 | 79.4 | 86.1 |
| | LEA * | 17.3 × 17.3 | 404 | 2003 | 6.7 | 79.4 | 86.1 |
| | AVE 1 * | 18 × 17.7 | 189 | 1544 | 4.8 | 54.8 | 70.73 |
| | AVE 2 * | 25 ha [2] | 500 | 143,280 | 4.0 | 56.1 | 85.6 |
| | LEA$_{SUP}$ ** | 17.3 × 17.3 | 403 | 2053 | 6.9 | 75.7 | 85.9 |
| | AVE 1 ** | 17.3 × 16.8 | 366 | 2109 | 7.0 | 76.5 | 86.7 |
| | AVE 2 ** | 30 ha [2] | 380 | 179,640 | 4.0 | 72.8 | 86.8 |
| Alfalfa | SUP * | 17.5 × 17.5 | 326 | 1923 | 6.3 | 77.0 | 84.1 |
| | LEA * | 17.5 × 17.5 | 325 | 1907 | 6.2 | 62.7 | 71.4 |
| | AVE 1 * | 17.5 × 17.5 | 308 | 1875 | 5.1 | 57.2 | 66.8 |
| | LEA ** | 33.4 ha [2] | 320 | 182,160 | 4.9 | 80.4 | 86.2 |
| Maize | SUP/LEA **$_1$ | 19 ha [2] | 250 | 97,200 | 4.0 | 82.3 | 86.2 |
| | SUP/LEA **$_2$ | 19 ha [2] | 250 | 115,560 | 4.0 | 84.3 | 89.9 |
| Sweet Maize | SUP/LEA ** | 20 ha [2] | - | 89,640 | 5.3 | 82.5 | 87.8 |

Notes: SUP: SUPROMED; LEA: leader farmer; AVE: average farmer; LEA$_{SUP}$: LEA using SUP platform; DU: distribution uniformity; CU: Christiansen's coefficient of uniformity. [1] Due to the lockdown imposed under the COVID 19 pandemic on March 15, these evaluations were not carried out. Estimated DU and CU values were included. [2] Area of the pivot irrigation system. [*] 2019–2020 campaign; [**] 2020–2021 campaign; [**$_1$] first evaluation of the centre pivot irrigation system; [**$_2$] second evaluation of the centre pivot irrigation system.

**Table 4.** Maize crop key performance indicators for SUPROMED (SUP) and leader (LEA) managements.

| | SUP (Tool) | LEA |
|---|------------|-----|
| Yield (kg ha$^{-1}$) | 15,142 | 14,304 |
| Fertilization (UN ha$^{-1}$) | 378 | 378 |
| Rainfall (mm) | 123 | 123 |
| ETc (mm) | 646 | 646 |
| Irrigation water (mm) | 622 | 773 |
| ET$_a$/ET$_m$ | 1 | 1 |
| Total percolation (mm) | 44 | 167 |
| Irrigation water percolation (mm) | 8 | 110.5 |
| Profitability (EUR ha$^{-1}$) | 1905 | 1489 |
| Irrigation water productivity (kg m$^{-3}$) | 2.4 | 1.9 |
| Irrigation water productivity (EUR m$^{-3}$) | 0.31 | 0.19 |
| Water footprint (m$^3$ kg$^{-1}$) | 0.58 | 0.61 |

As an example, we show the management of a maize crop irrigated by both the SUPROMED team and the farmer using a centre pivot irrigation system. Initially, the evaluation of the irrigation system showed a lack of uniformity in the centre pivot (Figure 10a). The blue line represents the gross water depth received at each point of the centre pivot radio. The blue and green horizontal lines represent the acceptable 15% of variation respect the average gross water depth applied by the irrigation system (black line). In the tracts where the black line surpassed these lines some emitters were substituted to improve the uniformity of the system (Figure 10b) (Table 3).

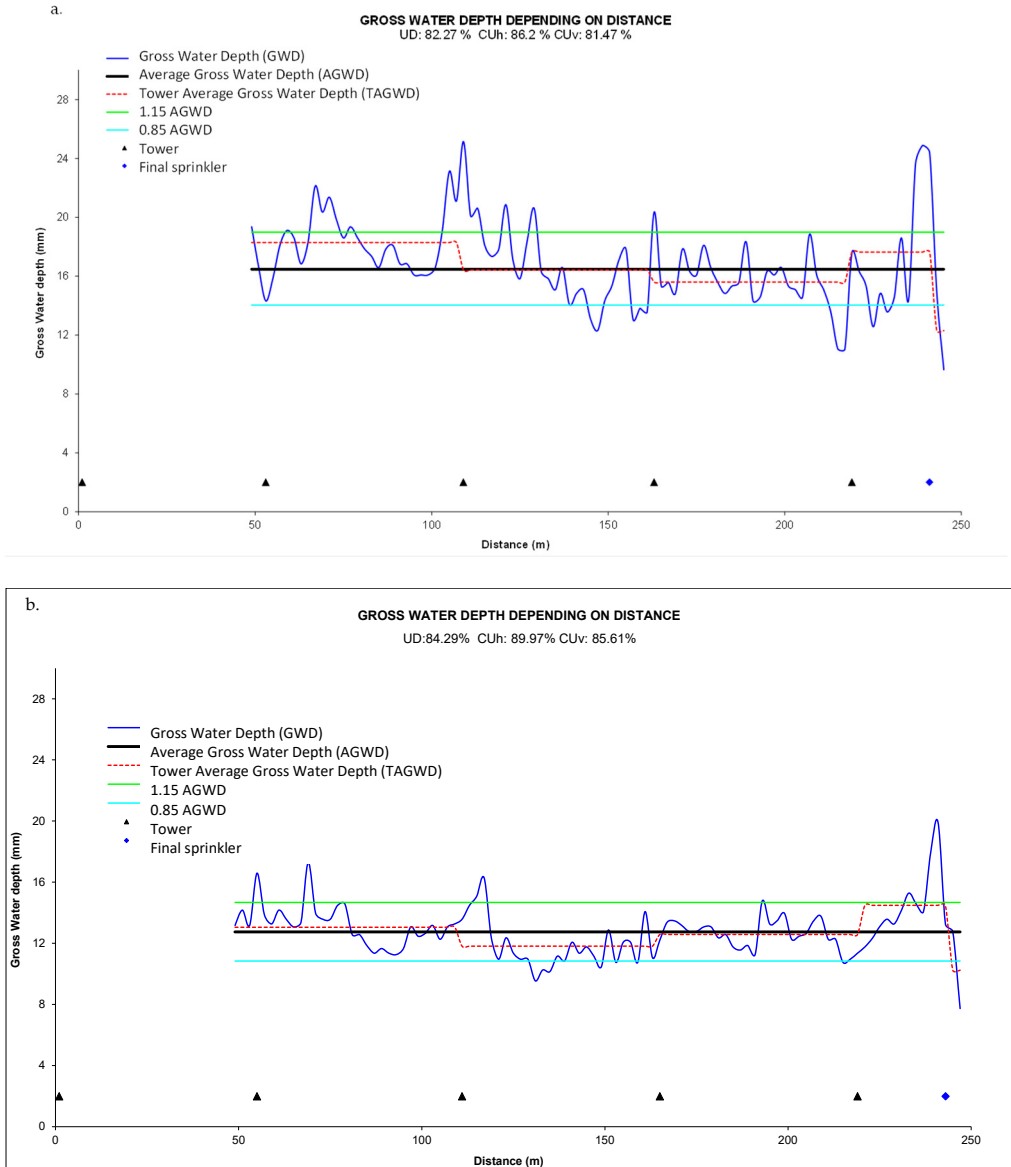

**Figure 10.** Results of the evaluation of a monitored maize crop pivot irrigation system. Image (**a**) before evaluation; (**b**) after evaluation. UD: distribution uniformity, $CU_H$: uniformity coefficient of Herman and Hein; $CU_V$: uniformity coefficient of variation [66].

During the growing season, the amount of irrigated water supplied by the farmer to the crop was 773 mm, while SUP supplied 622 mm, with this last value being consistent with other studies carried out in the area [16,61,63]. Thus, SUP resulted in irrigation water savings of 20%, in comparison with the farmer. Due to a slight positive difference in crop yield between SUP and farmer management (around 6%), this resulted in an increase in the irrigation water productivity for SUP management (around 21%) (Table 4) similar to that obtained by [67]. According to the tool, the total percolation depth of SUP was 74% lower than the farmer, with most of this being generated during the last stage of the crop cycle (ripening) when the irrigation requirements decreased, but the farmer maintained similar irrigation depths to those implemented during higher irrigation requirements (Figure 11b), while the model proposed decreasing the irrigation depths (Figure 11a). Compared to the farmer's management, SUP increased the irrigation water productivity by 26.3%, the profitability of the crop by 28% and reduced the total water footprint ($WF_{total}$) by around 5% (increasing $WF_{green}$, which implies a higher use of rainfall, and reducing $WF_{blue}$, which

means a lower use of irrigation water). Both values of $WF_{total}$ were lower than that obtained for the area [68]. These results increased the confidence of the farmer in the research team and in the model, spurring them to use this tool in future campaigns.

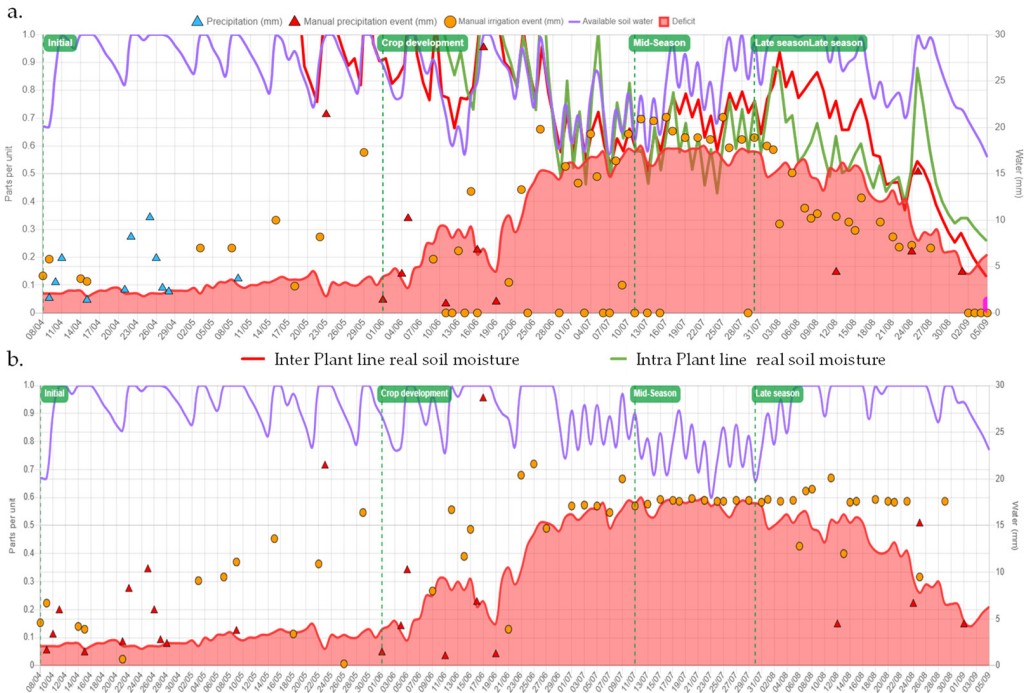

**Figure 11.** Progression of soil moisture simulated by "MOPECO irrigation scheduling" tool and comparison with the readings of the probes (Dill and Drop) installed in the plot of maize managed by SUP in Spain; (**a**) plot managed by SUPROMED team; (**b**) plot managed by the leader farmer without using the tool.

The simulation of the plot managed by SUP (Figure 11a) using the "MOPECO irrigation scheduling" tool and the plot managed by the farmer (simulated by the tool but not used by the farmer) (Figure 11b), shows the soil moisture progression in both plots represented by the available soil water line (purple) that moves between field capacity (AW = 1) and wilting point (AW = 0), the actual irrigation events (orange dots), the actual rainfall (blue triangles provided by the weather station and red triangles provided by the rain gauge installed at the plot) and the progression of the soil moisture readings measured by the probes installed in the SUP plot. Thus, in several moments of the cycle, the farmer exceeded the field capacity (when the AW line surpassed the top of the graph), causing percolation, mainly during the ripening stage. On the other hand, MOPECO maintained the total available water between field capacity and allowable depletion level (red area), thus generating a more intensive use of rainfall water and avoiding irrigation water percolation, as was corroborated by the soil moisture readings (Figure 11a).

### 3.3. Involving the Different Stakeholders in the Development of the Tool

During the second and third years, the use of the tool by farmers, assisted by the research team, facilitated feedback about the design and operability of the tool. The comments made by assistants in the meetings organized by SUPROMED, as well as the responses given by end users in the questionnaires, highlighted the following aspects to be improved with regard to the preliminary version of the model:

- Development of a mobile app to make the irrigation scheduling model more accessible to farmers that do not typically use computers but have a smartphone. Moreover, mobile phones allow users to have access to the tool in any place and at any time, which is also a great advantage. At this moment, the computer version is available on

the project website and the app is available for the Android operating system at the app store. The iOS version is under development at the time of writing this paper;

- Modifying the way weekly irrigation requirements were shown. Initially, this value was expressed in terms of water depth (mm), but many farmers prefer to receive this information in terms of irrigation hours. For this reason, the software was modified to show this output in both units (hours and mm) (Figure 12);

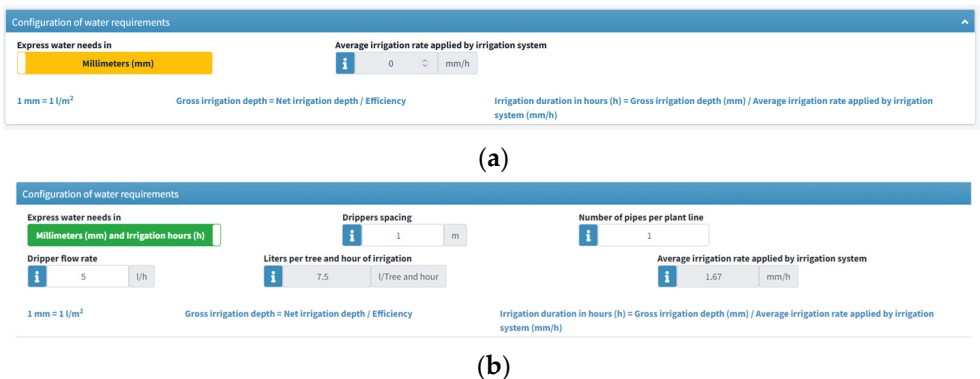

(**a**)

(**b**)

**Figure 12.** Irrigation requirements in mm (**a**) and in mm h$^{-1}$ (**b**).

- Implementation of the tool in other areas. Many farmers and technicians participating in the conferences and workshops organized by the SUPORMED members at the 3 demosites come from other areas and expressed concern about using this model in their areas. The tool was programmed taking this possibility into account. Thus, the adaptation to other areas and crops requires the involvement of an institution tasked with including the validated parameters required for the simulation of the crops (i.e., $K_c$ values, $K_c$ stages duration in GDD, and pictures of the phenological stages of the crops), and a proper network of weather stations providing the climatic data required to calculate the daily $ET_o$ in the area.

In addition, the questionnaires provided other useful information about the problems and necessities of the farmers at the three demosites. Broadly speaking, farmers highlight water scarcity as the most significant problem they must face, combined with the high cost of energy used for pumping water. Most of them know what a DSS tool is (79.2% in Spain, 80% in Lebanon and 10% in Tunisia), but only a low percentage have used one (52% in Spain, 20% in Lebanon and 3% in Tunisia), mainly focused on determining irrigation and fertilization requirements. In many cases, they abandoned the use of these DSS because they were not properly adapted to the necessities of the users or were difficult to use or the results difficult to interpret. Regarding the use of MOPECO, 60% of respondents in Lebanon and 70% in Spain found MOPECO easy to use. In the questionnaires, farmers were asked about their willingness to pay for DSS services. Thus, only if it is previously demonstrated that the tool increases the profitability of the farm would they pay for access to these tools and services. The total cost per year could range between EUR 7.90 ha$^{-1}$ in Tunisia, EUR 10 ha$^{-1}$ in Spain and EUR 33.00 ha$^{-1}$ in Lebanon. For that reason, it was decided that the irrigation scheduling tool would be provided for free, as a way to motivate end users in the demand of other services such as remote sensing advising, energy audits, evaluation of the irrigation systems, soil analysis, etc., that imply a higher cost.

The field visits, in situ evaluation of irrigation systems, and the sharing of experiences between farmers and researchers were highly useful in collecting information to improve the tool and to learn end users' opinions [25].

*3.4. Transferring the Tool to the Productive Sector*

The SUPROMED results were presented in 33 technical meetings, and 6 national and international congress organized by the research teams, receiving the attention of farmers, technicians, policy makers, researchers and irrigator communities. In addition, 25 mul-

titudinous training courses were organized, which were complemented with more than 40 individual training courses for farmers participating in the validation and adaptation processes of the tool. In total, more than 2200 persons attended all these dissemination activities (Table 5).

**Table 5.** Technical meetings, congresses and training courses organized during the SUPROMED project.

|                    | Spain | Lebanon | Tunisia | Others | Total |
| ------------------ | ----- | ------- | ------- | ------ | ----- |
| Technical meeting  | 8     | 5       | 20      | 0      | 33    |
| Congress           | 2     | 0       | 2       | 2      | 6     |
| Training course    | 3     | 2       | 19      | 1      | 25    |

Other dissemination activities for non-scientific individuals included the development of a website (www.supromed.eu (accessed on 15 January 2023)), a presence on social networks (Facebook, Twitter and LinkedIn), the publication of videos (11) on YouTube (https://www.youtube.com/channel/UCo11-inNByXq7-fvor4tlDA/videos (accessed on 15 January 2023)), printing brochures (3000 in 5 languages), posters (5), roll-up (1) and factsheets (4), and appearances in local press (34), radio (2) and TV programmes (4). In addition, manuals (7) and video tutorials (9) for the use of the different models and tools in the SUPROMED platform were developed (8 video tutorials for MOPECO irrigation scheduling). All these activities have increased the visibility of the project, as is demonstrated by more than 17,300 visits to the website, 164 likes on Facebook, 265 connections on Twitter and 374 on LinkedIn (data as of December 2022).

The use of "MOPECO irrigation scheduling" software is free. Nevertheless, users must register in order to save the data of their plots, crops and status of the irrigation scheduling. The number of users registered is about 300, mainly from Spain, because the IREY model [59] is the main tool in Tunisia for determining irrigation schedules on annual crops, while MOPECO irrigation scheduling is mainly used for tree crops (another option offered by the tool not commented on in this paper) in the three countries. In Lebanon, the number of users is low, mainly due to the low number of weather stations and calibrated crops (potato, wheat and maize) available in the area.

Other similar tools have been developed by private enterprises to assist farmers in irrigation scheduling [69,70]. However, farmers are unable to access them without paying for the product (irrigation system) or the service offered by the enterprise.

The interest of end users in SUPROMED methodologies has not escaped the public administrations that have understood the potential of implementing these tools in rural areas as a means to improve the quality of life (higher income and protection of the environment), promoting youth employment and integrating women into the management of farms by using methodologies that require trained professionals. Thus, the Rural Development Department of the regional government of Castilla–La Mancha (Spain) has signed a contract with the University of Castilla–La Mancha (one of the partners of the project) to implement certain SUPROMED tools and methodologies in the different irrigable areas of this region. Other institutions, such as IFAPA in Andalusia (Spain), have also shown interest in these models. Both regions aggregate more than 1,300,000 ha of irrigable land, which represents more than one third of the total irrigable area in Spain. In the same way, the Spanish Ministry of Agriculture has contacted us regarding teaching these methodologies in courses aimed at technicians. In Tunisia, a memorandum of understanding was signed between the Extension and Agriculture Training Agency (AVFA), the Institute of Agricultural Research and Higher Education (IRESA), and SUPROMED partners the National Institute for Research in Rural Engineering, Water and Forestry (INRGREF) Tunisia. IRESA, AVFA and INRGREF will continue using SUPROMED methodologies and tools and will initiate, stimulate and support joint projects that fit with the scope of SUPROMED project.

## 4. Conclusions

The great acceptation of the model by the productive sector in the demosite areas and the results obtained by the simplified version validate the methodology. The main advantage of the "MOPECO irrigation scheduling" tool is that it can improve the economic and environmental sustainability of farms, making more efficient use of irrigation water (28% and 26% higher profitability and water productivity, respectively, and 5% lower water footprint for maize). In addition, the model requires limited information, training, and equipment with regards to the version for researchers.

The identification of the problems farmers have to face (low availability of water, low profitability of farms, global warming, lack of irrigation advisory services) was carried out based on the research teams' deep knowledge of the sector, close contact and collaboration with stakeholders from many years before the development of the tool, and direct consultation with end users through personal interviews and questionnaires.

The integration of the stakeholders in the design of the tool made it possible to determine the plot, crop, soil and irrigation system parameters to be inserted in the model by the user, as well as adapting how the results are shown on the screen to their level of training and needs.

The validation of the tool under actual management conditions made it possible to first demonstrate to selected and influential farmers, and then to other technicians and farmers through workshops and conferences, the ability of the model to help them improve the irrigation scheduling of their farms.

The ambitious project dissemination programme, its aims, and the results obtained through conferences, videos, tutorials on the use of the models, and appearances in the media and on social networks allowed us to reach a large number of potential users, which may increase through collaboration agreements signed with public institutions interested in expanding the scope of application of the tool to other irrigable areas.

As expected, the results provided by the "MOPECO irrigation scheduling" tool are less accurate than those offered by the full version of MOPECO for researchers, which also offers many more simulation options. However, thanks to its high level of scientific content and validation under rigorous field tests, we were able to obtain a simplified version that is sufficiently accurate for commercial use.

**Author Contributions:** Conceptualization, A.D. and J.A.M.-L. methodology, A.D.; validation, J.A.M.-L., formal analysis, A.D.; investigation, A.D., J.A.M.-L., F.K., H.A. and R.N.; resources, A.D.; data curation, J.A.M.-L.; writing—original draft preparation, A.D. and J.A.M.-L.; writing—review and editing, A.D., J.A.M.-L., F.K., H.A., R.N. and M.O.; visualization, M.O.; supervision, A.D.; project administration, A.D.; funding acquisition, A.D. All authors have read and agreed to the published version of the manuscript.

**Funding:** This research was carried out within the European project SUPROMED "GA-1813" funded by PRIMA.

**Data Availability Statement:** Not applicable.

**Acknowledgments:** The authors thank the farmers, irrigators communities and farmers associations participating in this research for their support in the tasks and actions performed during the three years of the project. In addition, we also thank META-Group (www.meta-group.com (accessed on 8 January 2023)) for their support in the use of the methodology for reaching a success development and transference of a model to the productive sector.

**Conflicts of Interest:** The authors declare no conflict of interest. The funders had no role in the design of the study; in the collection, analyses, or interpretation of data; in the writing of the manuscript; or in the decision to publish the results.

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
