# Peer review of "Adaptation of a Scientific Decision Support System to the Productive Sector—A Case Study: MOPECO Irrigation Scheduling Model for Annual Crops"

_water, doi:10.3390/w15091691_

Round 1

Reviewer 1 Report

This manuscript addresses the dissemination and use of a previously developed decision support (DSS) system (MOPECO) for irrigation scheduling in arid and semiarid regions. It is written is good English and the manuscript structure seems appropriate.

However, there seems to me that it may lack some novelty and significance for the scientific community. It is mostly concerned with addressing the use of the developed DSS by technicians, farmers, advisory enterprises, regional or national governmental entities, rather than in soundly demonstrating the accuracy of the proposed model in the actual real situations.

Therefore, I believe that this manuscript would benefit by an in-depth analysis on the comparison of the results obtained by the model and actual field readings, which seem shortcoming to me. Other matters in the proposed manuscript that I believe should be improved are presented in the attached review file.

Author Response

The depth analysis of the comparison between the model and field readings is shown with more accuracy in other articles included in the references (numbers 54 and 55). 

Ok. Thanks. The improvements presented in the attached review file have been considered in the new version of the manuscript.

Reviewer 2 Report

Comments

The main objective of the study conducted by Domínguez et al. entitled, “Adaptation of a Scientific decision support system to the Productive Sector. A case study: MOPECO irrigation scheduling Model for annual crops” was to maximize the profitability of irrigated farms in water-scarce regions as described in the abstract but the author and to transform a tool for scientific use into one for end users to achieve an efficient irrigation scheduling at farm level. The authors produced a nice user interface/software. However, the presentation of results needs some improvements, which are described below:

Line 86-87. Suppose MOPECO needs many variables and crop parameters as other simulation models. What is the advantage of developing this software?

Figure 1. Add coordinates, scale, and north arrow

Line 156. First, explain the abbreviation, then use it for example. Ym, Kc, and Ky were not explained first.

In section 2.1, you could further sub-classify the headings. For example, 2.1.1 for Spanish demo sites.  

Line 391-392, rephrase it to a meaningful sentence and enlist/specify clearly the items used in the software for farmers.

The authors have explained the procedure in the flow diagram. Better to include the interface figures (3-7) of the software in the flow diagram making the manuscript a research article rather than a user interface manual.

Figure 9 caption needs rephrasing. Farmer Vs SUPROMED management ratio was never defined before in the article. One could not understand the y-axis of the figure. The item in the legend, i.e., Ig never used before in the text.

Improve the resolution of all Figures. Explain all the figures in the text. For example, you have explained Figure 10 in lines 516-519 only, which needs more explanation to increase the readability. The legends of the figures are not explained in the text. Spelling mistakes in the legend, i.e., Collected depth

Figure 11 is the main part of your study. Improve its resolution and better plot the results using Excel and other available software. I could not see the gross irrigation points in the figure, but it is presented in the legends.

Improve the conclusion based on the objective. Line 677-679 is a redundant statement with no proper meaning.

If possible, please explain the difference in input and output of MOPECO compared to the aqua crop model. Why MOPECCO be proffered over aqua crop?

Author Response

  • Line 86-87. Suppose MOPECO needs many variables and crop parameters as other simulation models. What is the advantage of developing this software?

Ok. Lines 99-100, 437-443

  • Figure 1. Add coordinates, scale, and north arrow

Ok.

  • Line 156. First, explain the abbreviation, then use it for example. Ym, Kc, and Ky were not explained first.

Ok. Lines 178 (Kc), 182 (Ym) and 184 (Ky).

  • In section 2.1, you could further sub-classify the headings. For example, 2.1.1 for Spanish demo sites.  

Ok: Lines 125, 138 and 149.

  • Line 391-392, rephrase it to a meaningful sentence and enlist/specify clearly the items used in the software for farmers.

Ok. Lines 437-443.

  • The authors have explained the procedure in the flow diagram. Better to include the interface figures (3-7) of the software in the flow diagram making the manuscript a research article rather than a user interface manual.

We have modified figure 2 to clarify the simplification carried out with regards the MOPECO for researchers version. Nevertheless, we have preferred to maintain the interface figures in the same position due to inserting them in figure 2, which corresponds to material and methods section instead of results section may cause confusion to readers. Anyway, we understand the comment of the reviewer and we assume the nature of this paper is in the border between research and transference, being the methodology used for transferring the model and its validation in real farms the factors that allow to consider this paper as a research work.

  • Figure 9 caption needs rephrasing. Farmer Vs SUPROMED management ratio was never defined before in the article. One could not understand the y-axis of the figure. The item in the legend, i.e., Ig never used before in the text.

Ok. Figure 9 caption and Lines 605-612.

  • Improve the resolution of all Figures. Explain all the figures in the text. For example, you have explained Figure 10 in lines 516-519 only, which needs more explanation to increase the readability. The legends of the figures are not explained in the text. Spelling mistakes in the legend, i.e., Collected depth

The resolution of the figures has been improved, we will change the size of the letter if required by the journal editing service; Lines 605-612.

  • Figure 11 is the main part of your study. Improve its resolution and better plot the results using Excel and other available software. I could not see the gross irrigation points in the figure, but it is presented in the legends.

Ok. It´s important to show that the pictures showed in Fig.11 are the real pictures that the MOPECO irrigation scheduling shows to end users. Letter size in the figures will be increased by demand of the editing service of the journal if the size is not appropriate to the published version of the document. As suggested, the resolution has been improved.

  • Improve the conclusion based on the objective. Line 677-679 is a redundant statement with no proper meaning.

Ok. Lines 792-798.

If possible, please explain the difference in input and output of MOPECO compared to the aqua crop model. Why MOPECCO be proffered over aqua crop?

This affirmation is based on the reference nº 12.

Reviewer 3 Report

The current manuscript entitled “Adaptation of a scientific decision support system to the productive sector. A case study: MOPECO irrigation scheduling model for annual crops” by Domínguez et al. presents an interesting case study on the adaptation of a scientific decision support system for use in the productive sector. The paper highlights the successful adaptation of the MOPECO irrigation scheduling model for annual crops and the challenges faced during the adaptation process. After a careful review, I found this manuscript interesting. However, the manuscript requires extensive editing in terms of English language and typographical corrections. Also, there are a few sections where the paper could be improved. Firstly, the authors could provide more details on the data used in the study, such as the metholdogy of the data and the sampling practice. Secondly, the paper could benefit from more discussion on the limitations and future directions of the adapted MOPECO model. Overall, my suggestion is a major revision. My specific comments are:

1.      The title of the paper is very confusing, please rewrite it completely.

2.      The abstract is not written properly and scientifically. It should be re-written in the following order: Research problem, solution, aims/objectives of this study, research design, major outcome, the relevance of the outcomes, and overall contribution of results to current knowledge.

3.      Line 25: approach adopted to adapt? Correct it.

4.      The abstract should include major numerical findings.

5.      The geocoordinate information of demo sites should be given.

6.      The order of sections and subsections is hard to understand. Please rearrange and number all of them carefully.

7.      The multiplication sign should be corrected (Table 3 and other text parts).

8.      The conclusion section should be reduced to <250 words.

9.      Several typos and English errors must be corrected, I can’t mention them all here because they are too many.

Author Response

3.1. The title of the paper is very confusing, please rewrite it completely.

Dear reviewer, in our opinion the title reflects the aim of the paper and no other reviewers have suggested to modify it, and due to you have not indicated what is the problem, we have decided to keep the original one.

3.2. The abstract is not written properly and scientifically. It should be re-written in the following order: Research problem, solution, aims/objectives of this study, research design, major outcome, the relevance of the outcomes, and overall contribution of results to current knowledge.

Ok, the abstract has been modified. Lines 16-35.

3.3. Line 25: approach adopted to adapt? Correct it.

Ok. Line 25

3.4. The abstract should include major numerical findings.

Ok. Line 29.

3.5. The geocoordinate information of demo sites should be given.

Ok. Figure 1.

3.6. The order of sections and subsections is hard to understand. Please rearrange and number all of them carefully.

Ok. Lines 125, 138 and 149.

3.7. The multiplication sign should be corrected (Table 3 and other text parts).

Ok. Corrected. Table 3

3.8. The conclusion section should be reduced to <250 words.

We have summarized the conclusions but the total number of words is still higher (we have consulted the instructions for authors and there is not a limitation in the total number of words).

3.9. Several typos and English errors must be corrected, I can’t mention them all here because they are too many.

Ok. The English has been reviewed by a native English speaker (a certificate is available)

Reviewer 4 Report

The process of adapting, validating, and transferring the "MOPECO irrigation planning" model described in this article for farmers and technicians is laid out. The main advantage of this tool is that it can improve the economic and environmental sustainability of farms by enabling more efficient use of irrigation water. It is an important study in that the model uses soil, crop, and climate data and can be transferred to irrigation systems to increase plant productivity.

The resolution should be increased to understand the figures in the article better.

In the summary section, it would be good to give numerical results for applying the model in three locations.

The problems encountered in the application and recommendations section should be detailed in the conclusion and recommendations section.

In the study, evaluating the program proportionally in terms of efficiency before and after the application would be good.

Author Response

4.1. The resolution should be increased to understand the figures in the article better.

Ok. The resolution of all the figures has been improved.

4.2. In the summary section, it would be good to give numerical results for applying the model in three locations.

Ok. Line 29.

4.3. The problems encountered in the application and recommendations section should be detailed in the conclusion and recommendations section.

Ok, Lines 779-780.

4.4. In the study, evaluating the program proportionally in terms of efficiency before and after the application would be good.

In Table 4 an example for maize crop is showed. LEA management represents the traditional management (before the use of the MOPECO irrigation scheduling). The evaluation proposed is showed in other articles (references 54 and 55).

Round 2

Reviewer 1 Report

Given the modifications performed, I believe that this manuscript is now able to be considered for publication, though some issues with the novelty and significance for the scientific community still remain.

Reviewer 2 Report

Figure 10. Please check/correct the spelling in the legends. i.e., Depth instead of Depht

Fig. 8. Improve the resolution. It is difficult to read the text in the legends and table.  

Author Response

2.1. Figure 10. Please check/correct the spelling in the legends. i.e., Depth instead of Depht.

Ok, thank you.

2.2. Fig. 8. Improve the resolution. It is difficult to read the text in the legends and table.

Ok. We understand and appreciate the worries of the reviewer. It´s important to highlight Fig.8 is a screenshot of the results that MOPECO irrigation scheduling shows to end users. Therefore we prefer not to change the size of the letter or cut the figure to enlarge a portion. As suggested, the resolution of the figure has been improved taking it from a FHD screen. This resolution makes it possible to appreciate the details of the figure if zooming on the figure when reading the digital version of the paper (of course, it is impossible if the reader reads the printed version), which is the most common option nowadays. Anyway, if the editing service of the journal ask for that, we can turn the figure and increase its size up to the size of the paper sheet.

We appreciate for your efforts to improve the quality of our paper.

Reviewer 3 Report

The authors have answered my queries and corrected manuscript as per comments. I have no further comment.

Author Response

The authors have answered my queries and corrected manuscript as per comments. I have no further comment.

We appreciate for your efforts to improve the quality of our paper.

Reviewer 4 Report

.

Author Response

We appreciate your efforts to improve the quality of our paper.